# Physics-Informed Implicit Representations of Equilibrium Network Flows

**Kevin D. Smith**[*]     **Francesco Seccamonte**[*]     **Ananthram Swami**[†]     **Francesco Bullo**[*]

## Abstract

Flow networks are ubiquitous in natural and engineered systems, and in order to understand and manage these networks, one must quantify the flow of commodities across their edges. This paper considers the estimation problem of predicting unlabeled edge flows from nodal supply and demand. We propose an implicit neural network layer that incorporates two fundamental physical laws: conservation of mass, and the existence of a constitutive relationship between edge flows and nodal states (e.g., Ohm's law). Computing the edge flows from these two laws is a nonlinear inverse problem, which our layer solves efficiently with a specialized contraction mapping. Using implicit differentiation to compute the solution's gradients, our model is able to learn the constitutive relationship within a semi-supervised framework. We demonstrate that our approach can accurately predict edge flows in AC power networks and water distribution systems.

## 1 Introduction

Network flows are a fundamental aspect of modern society, from traffic and communication networks to power and water distribution systems. Many critical infrastructures are well-modeled as graphs, with edges that transport vital commodities [1]. Beyond infrastructure, network flows are also central to models in epidemiology, ecology, medicine, and chemical networks. Their dynamics have been well-studied in compartmental systems theory [2]. Given the prevalence of flow networks in natural and engineered systems, predicting flows in these networks is an important learning task that may facilitate monitoring, control, optimization, and protection of these networks.

While domain-specific tools to predict network flows have been around for a while, the machine learning community has only recently taken an interest in general-purpose models for network flows. [3] predicts edge flows from partial measurements by making a smoothing assumption, i.e., by minimizing nodal flow divergence. [4] improves on this approach by adding a trainable regularizer that can incorporate side information. Both of these approaches are centered on a notion of approximate conservation, i.e., that the net inflow to each node should be near zero. Since conservation of mass is a universal constraint on network flows, imposing this conservation law is an important step toward embedding physics into the model.

But the conservation law alone is not enough to uniquely determine flow, which is why both [3] and [4] rely on heuristic regularizers to select the "best" conservation-respecting flow. In fact, physical networks are often governed by a *pair* of physical laws: the conservation law, and a *constitutive relationship*, which specifies the magnitude and direction of each edge flow based on "effort" variables at each incident node (e.g., pressure or voltage). For example, in DC circuits, currents are conserved according to Kirchoff's current law, and Ohm's law is the constitutive relationship that relates current flows to nodal potentials. The conservation law and the constitutive relationship together define the unique edge flows (and nodal efforts).

---

[*]Center for Control, Dynamical Systems, and Computation, University of California, Santa Barbara, CA 93106 USA, {`kevinsmith, fseccamonte, bullo`}`@ucsb.edu`

[†]U.S. Army Research Laboratory, Adelphi, MD 20783 USA, `ananthram.swami.civ@army.mil`

36th Conference on Neural Information Processing Systems (NeurIPS 2022).

## 1.1 Contributions

This paper proposes a model for network flows that embeds both the conservation law and existence of a constitutive relationship. Our model, which we call an *Implicit Flow Network* (IFN), predicts each edge flow using a trainable nonlinear function of latent nodal variables. These latent variables are constrained to a manifold wherein the conservation law is satisfied. In addition to introducing IFN, we offer the following contributions: (i) a contraction algorithm that is able to both evaluate the IFN layer and backpropagate gradients through it, (ii) an explicit upper bound on the number of iterations required by this algorithm, (iii) a rigorous theoretical comparison between IFN and the state-of-the-art flow estimation methods in [3, 4], and (iv) numerical experiments from several AC power networks and water distribution systems that indicate IFN can significantly outperform these baselines on the flow estimation task. Additionally, because IFN requires a nonlinearity with a constrained slope, we provide (v) a novel "derivative-constrained perceptron", which is essentially a trainable activation function with upper and lower bounds on its slope.

## 1.2 Related Work

**Network Flow Estimation** Flows on graphs are a classical topic in computer science [5], and flow forecasting has long been studied in specific domains like traffic [6], but interest in the flow estimation task from a machine learning perspective appears to be relatively recent. Deep learning algorithms have been used to predict traffic flows [7, 8] and power flows [9], but [3] and [4] appear to be the first papers to propose methods for domain-agnostic flow prediction, based on the notion of divergence minimization.

**Implicit Neural Networks** IFN belongs to a growing class of models called implicit neural networks, which do not explicitly state the output of the model; rather, they describe a desired relationship between the model's inputs and outputs. In the prevailing implicit framework, the output is defined as a fixed point of a trainable perceptron. This approach was introduced in [10] as a "deep equilibrium network". Subsequent work has developed new frameworks for ensuring the existence of the fixed point and computing it [11, 12, 13, 14, 15]. Other types of implicit neural networks include neural ODEs [16] and layers that solve convex optimization problems [17] and Nash equilibria [18].

**Graph Neural Networks** Graph neural networks (GNN) are a diverse family of models for network-related learning tasks that incorporate graph structure directly into the model. GNNs can typically be classified into three types, in increasing order of generality [19, §5.3]: convolutional models [20, 21], attentional models [22], and message-passing models [23, 24]. Recently, [25] proposed an implicit graph convolutional network. Analogously, IFN can be interpreted as an implicit message-passing GNN, with flows serving as messages and latent nodal variables acting as an embedding.

## 1.3 Preliminaries and Notation

Given a directed graph $G = (\mathcal{V}, \mathcal{E})$, the signed incidence matrix $B \in \{-1, 0, 1\}^{|\mathcal{V}| \times |\mathcal{E}|}$ is the matrix with entries

$$B_{i,e} = \begin{cases} 1, & i \text{ is the head of } e \\ -1, & i \text{ is the tail of } e \\ 0, & \text{else} \end{cases} , \quad \forall i \in \mathcal{V} \text{ and } e \in \mathcal{E}$$

For an undirected graph, the signed incidence matrix is obtained by assigning an aribtrary orientation to each edge. For each $i \in \mathcal{V}$, let $\mathcal{N}_{\text{in}}(i), \mathcal{N}_{\text{out}}(i) \subset \mathcal{V}$ be the in-neighbors and out-neighbors of $i$.

Given a vector $x \in \mathbb{R}^n$, we use the notation $[x]$ to denote the diagonal matrix $\text{diag}(x) \in \mathbb{R}^{n \times n}$. Where such notation would be unclear (e.g., may be confused with brackets to indicate order of operations), we fall back on the $\text{diag}(\cdot)$ notation. We write $x^\perp$ to refer to the vector space that is orthogonal to $x$, i.e., the space $\{x' \in \mathbb{R}^n : x^\top x' = 0\}$. Given a positive definite diagonal matrix $D \in \mathbb{R}^{n \times n}$, we write $||x||_{2,D}$ to represent the weighted 2-norm $||D^{\frac{1}{2}} x||_2$. Given any matrix $M$, $M_i$ is the $i$th column vector of $M$, and $M^{(j)}$ is the transpose of the $j$th row vector.

| Flow | Nodal Variable | $h(y) =$ |
|---|---|---|
| DC Current | Voltage | $y$ |
| DC Power | Voltage | $y^2$ |
| AC Power (lossless) | Voltage Angle | $\sin(y)$ |
| Water Flow Rate | Hydraulic Head | $\text{sign}(y)|y|^{0.54}$ |
| Mechanical Force Networks | Position | $y$ |

Table 1: Examples of physical flow networks and their constitutive relationships.

## 2 Implicit Flow Networks

IFN is inspired by the physics of network systems. In many physical networks, nodes "communicate" through the exchange of a commodity, like power, water, or force, which can be represented as edge flows. Flows obey a conservation law: for all $i \in \mathcal{V}$,

$$0 = u_i + \overbrace{\sum_{j \in \mathcal{N}_{\text{in}}(i)} f_{(i,j)}}^{\text{net inflow}} - \overbrace{\sum_{j' \in \mathcal{N}_{\text{out}}(i)} f_{(i,j')}}^{\text{net outflow}}, \tag{1}$$

where $u \in \mathbb{R}^{|\mathcal{V}|}$ are nodal inflows from outside the network, and $f \in \mathbb{R}^{|\mathcal{E}|}$ are the edge flows. Furthermore, the flows are related to nodal variables through a constitutive relationship (CR); there is some strictly increasing function $h$ such that, for all $(i, j) \in \mathcal{E}$,

$$f_{(i,j)} = a_{(i,j)} h(x_i - x_j), \tag{2}$$

where $a \in \mathbb{R}^{|\mathcal{E}|}$ are edge weights and $x \in \mathbb{R}^{|\mathcal{V}|}$ are nodal "efforts" or "potentials." For example, in DC power networks, the CR is Ohm's law $f_{(i,j)} = r_{(i,j)}^{-1}(x_i - x_j)$, where $r$ are resistances and $x$ are voltages. In lossless AC networks, the CR is the active power flow equation $f_{(i,j)} = a_{(i,j)} \sin(x_i - x_j)$, where the edge weights are a function of line parameters and $x$ are voltage angles [26, §6.4]. In water distribution systems, the CR is the Hazen-Williams formula [27, Sec. 8.15]. Table 1 lists several flow networks, the physical interpretation of the effort variables $x$, and the flow function $h$.

We propose IFN as a layer that predicts edge flows based on these two physical laws—conservation and the existence of a CR:

**Definition 2.1** (Implicit Flow Network). An *implicit flow network* (IFN) is a module with the following components:

1. fixed parameters $0 < d_{\min} \le d_{\max}$,

2. trainable parameters $\theta \in \mathbb{R}^r$ for some $r$, and

3. a family of differentiable functions $h_\theta : \mathbb{R} \to \mathbb{R}$ such that $d_{\min} \le h'_\theta(y) \le d_{\max}$ for all $y \in \mathbb{R}$ and $\theta \in \mathbb{R}^r$, which we call *flow functions*.

The module requires each of the following inputs:

1. a weighted, connected, undirected graph $G = (\mathcal{V}, \mathcal{E}, a)$ with edge weights $a \in \mathbb{R}_{>0}^{|\mathcal{E}|}$, and

2. a supply / demand vector $u \in \mathbb{R}^{|\mathcal{V}|}$ such that $\sum_{i \in \mathcal{V}} u_i = 0$.

The module outputs the unique vector $f \in \mathbb{R}^{|\mathcal{E}|}$ for which there exists $x \in \mathbb{R}^{|\mathcal{V}|}$ such that

$$Bf = u \tag{3}$$

$$f = [a]h_\theta(B^\top x) \tag{4}$$

where $B \in \{-1, 0, 1\}^{|\mathcal{V}| \times |\mathcal{E}|}$ is the signed incidence matrix of $G$, and $h_\theta$ is applied element-wise. We use the notation $\text{FN}_{h,\theta}(G, u)$ to represent the solution $f$ given inputs $G$ and $u$, flow functions $h$, and parameters $\theta$.

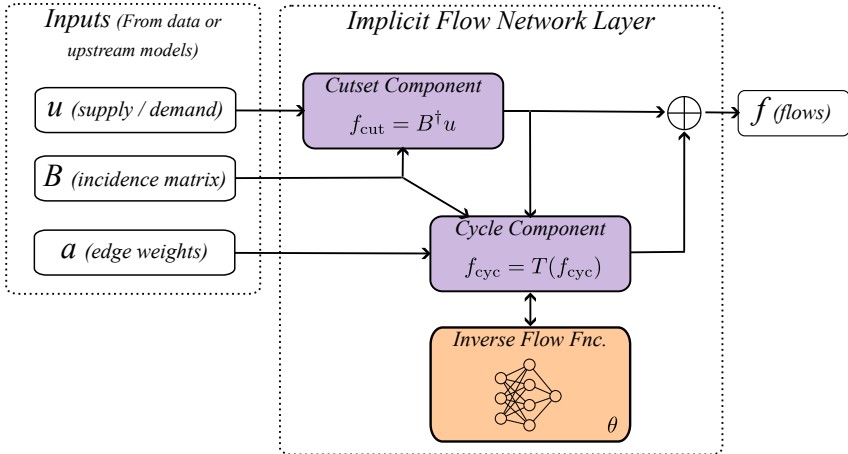

Figure 1: Diagram of the IFN. Inputs are the supply / demand vector $u$, incidence matrix $B$, and edges weights $a$, which are either known or output from upstream models. The IFN layer separately computes the cutset component and cycle component of the flows, with a trainable model for the inverse of the flow function in the CR. These components are summed and output as the flow prediction, for downstream use.

We will prove that IFNs are well-posed in Theorem 2.2. Note that (3) and (4) are just vectorized statements of the conservation law (1) and the CR (2), so these two physical laws directly define the output. The IFN's only trainable component is its flow function, parameterized by $\theta$. In practice, we will only make calls to the *inverse* of the flow function when evaluating and backpropagating through IFN layers, so it is convenient to learn the inverse flow function directly.

We emphasize that IFNs are layers that can be situated in more complex architectures, with other models upstream estimating the supply / demand vector, edge weights, or even the topology. For example, in power systems, demand forecasting is a very well-studied problem [28, 29], and one can solve the economic dispatch problem to forecast power generation at each node [30], collectively leading to an estimate of the supply / demand vector.

## 2.1 Evaluating the Implicit Flow Network

Our approach to evaluating the implicit flow network is adapted from [31] and is illustrated in Figure 1. Any undirected graph $G$ induces a direct decomposition of the edge flow space $\mathbb{R}^{|\mathcal{E}|}$: given the incidence matrix $B \in \{-1, 0, 1\}^{|\mathcal{V}| \times |\mathcal{E}|}$, the *cycle space* $\ker(B)$ and *cutset space* $\mathrm{Img}(B^\top)$ are orthogonal, and $\mathbb{R}^{|\mathcal{E}|} = \ker(B) \oplus \mathrm{Img}(B^\top)$. We refer the reader to [32, §9.4] for a primer on cycle and cutset spaces. Accordingly, we decompose the vector $f = \mathrm{FN}_{h,\theta}(G, u)$ as $f = f_{\mathrm{cyc}} + f_{\mathrm{cut}}$, where $f_{\mathrm{cyc}} \in \ker(B)$ and $f_{\mathrm{cut}} \in \mathrm{Img}(B^\top)$. The cutset component is readily determined from (3), since $Bf = Bf_{\mathrm{cut}} = u$ implies that $f_{\mathrm{cut}} = B^\dagger u$. Then we must analyze (4) to solve for $f_{\mathrm{cyc}}$. Define a *cycle projection matrix* $P \in \mathbb{R}^{m \times m}$ as the oblique projection onto $\ker(B)$ parallel to $\mathrm{Img}([a]B^\top)$:

$$P = I_m - [a]B^\top \left(B[a]B^\top\right)^\dagger B \tag{5}$$

Based on this projection, we define a map $T : \ker(B) \to \ker(B)$ for all $f_{\mathrm{cyc}} \in \ker(B)$ by

$$T(f_{\mathrm{cyc}}) = P\left(f_{\mathrm{cyc}} - d_{\min}[a]h_\theta^{-1}([a]^{-1}f_{\mathrm{cyc}} + [a]^{-1}B^\dagger u)\right) \tag{6}$$

We can show that $f_{\mathrm{cyc}}$ is the unique fixed point of $T$, and that $T$ is a contraction mapping, leading to a simple algorithm to compute this fixed point.

**Theorem 2.2** (Properties of $T$). *Consider an implicit flow network with parameters $d_{\min}$, $d_{\max}$, and $\theta$, with flow functions $h_\theta$. Suppose that the inputs $G = (\mathcal{V}, \mathcal{E}, a)$ and $u \in \mathbb{1}_{|\mathcal{V}|}^\perp$ are given, and let $B \in \{-1, 0, 1\}^{|\mathcal{V}| \times |\mathcal{E}|}$ be the signed incidence matrix of $G$. The following are true:*

1. *$T$ is a contraction mapping with respect to $||\cdot||_{2,[a]^{-1}}$, with Lipschitz constant*

$$\mathrm{Lip}(T) \leq 1 - \frac{d_{\min}}{d_{\max}},$$

2. *the sequence of iterates $f_{\mathrm{cyc}}^{(k+1)} = T(f_{\mathrm{cyc}}^{(k)})$ starting from any initial condition $f_{\mathrm{cyc}}^{(0)} \in \ker(B)$ converges to a unique fixed point $f_{\mathrm{cyc}}$,*

3. *the output of the implicit flow network is unique and given by*

$$\mathrm{FN}_{h,\theta}(G, u) = f_{\mathrm{cyc}} + B^\dagger u \tag{7}$$

*Consequently, IFN is well-posed.*

Theorem 2.2 provides a simple algorithm for computing the IFN output $f$: pick any $f_{\mathrm{cyc}}^{(0)} \in \ker(B)$, repeatedly apply the map $T$ until approximate convergence, then add $B^\dagger u$. Some care is required when implementing this map. Since $P$ is a dense matrix with $|\mathcal{E}|^2$ entries, it is undesirable to explicitly construct the cycle space projection matrix for large networks. Instead, in order to project a vector $v \in \mathbb{R}^{|\mathcal{E}|}$, we can use the fact that

$$w \triangleq (B[a]B^\top)^\dagger Bv = \underset{w \in \mathbb{R}^n}{\mathrm{argmin}} \left\{ ||B[a]B^\top w - Bv||_2 \right\}$$

so the projection is evaluated as $Pv = v - [a]B^\top w$. Using this method of projection to implement $T$, the fixed point iteration to compute $\mathrm{FN}_{h,\theta}(G, u)$ is stated in Algorithm 1.

---

**Algorithm 1** Evaluating the implicit flow network.

---

1: $B \leftarrow$ signed incidence matrix of $G$
2: $f_{\mathrm{cut}} \leftarrow \mathrm{argmin}_{f_{\mathrm{cut}} \in \mathbb{R}^m} \{||Bf_{\mathrm{cut}} - u||_2\}$
3: $f_{\mathrm{cyc}} \leftarrow \mathbb{0}_m$
4: $\Delta f_{\mathrm{cyc}} \leftarrow \infty \mathbb{1}_m$
5: **while** $||\Delta f_{\mathrm{cyc}}||_{2,[a]^{-1}} > \epsilon$ **do**
6:     $v \leftarrow d_{\min}[a]h_\theta^{-1} \left( [a]^{-1} f_{\mathrm{cyc}} + [a]^{-1} f_{\mathrm{cut}} \right)$
7:     $w \leftarrow \mathrm{argmin}_{w \in \mathbb{R}^n} \left\{ ||B[a]B^\top w - Bv||_2 \right\}$
8:     $\Delta f_{\mathrm{cyc}} \leftarrow v - [a]B^\top w$
9:     $f_{\mathrm{cyc}} \leftarrow f_{\mathrm{cyc}} - \Delta f_{\mathrm{cyc}}$
10: **end while**
11: $f \leftarrow f_{\mathrm{cyc}} + f_{\mathrm{cut}}$
12: **return** $f$

---

**Theorem 2.3** (Implicit Flow Networks, Forward Pass). *Consider an implicit flow network with parameters $d_{\min}$, $d_{\max}$, and $\theta$, with flow functions $h_\theta$. Suppose that the inputs $G = (\mathcal{V}, \mathcal{E}, a)$ and $u \in \mathbb{1}_{|\mathcal{V}|}^\perp$ are given, and let $B \in \{-1, 0, 1\}^{|\mathcal{V}| \times |\mathcal{E}|}$ be the signed incidence matrix of $G$. The following are true of Algorithm 1, with a tolerance of $\epsilon > 0$:*

1. *for each iteration $k = 1, 2, \ldots$ of the loop, let $f_{\mathrm{cyc}}^{(k)}$ represent the new value of $f_{\mathrm{cyc}}$ defined on line 9; and let $f_{\mathrm{cyc}}^{(0)} = \mathbb{0}_m$. Then*

$$f_{\mathrm{cyc}}^{(k+1)} = T(f_{\mathrm{cyc}}^{(k)}), \quad \forall k \geq 0;$$

2. *the algorithm converges with at most $k^*$ iterations of the while loop, where*

$$k^* = 1 + \frac{\log\left(d_{\min}^{-1}\rho^{-1}\epsilon\right)}{\log\left(1 - \frac{d_{\min}}{d_{\max}}\right)} \tag{8}$$

*and $\rho = ||[a]^{\frac{1}{2}} h_\theta^{-1}([a]^{-1} B^\dagger u)||_2$; and*

3. *the algorithm returns $f \in \mathbb{R}^{|\mathcal{E}|}$, where*

$$||f - \mathrm{FN}_{h,\theta}(G, u)||_{2,[a]^{-1}} \leq \left( \frac{d_{\max} - d_{\min}}{d_{\min}} \right) \epsilon \tag{9}$$

If evaluating $h_\theta^{-1}$ is sufficiently simple, then the most expensive step in the iteration is solving the ordinary least squares problem on line 7. Using a general-purpose solver, the complexity of this

operation is roughly $O(|\mathcal{V}|^3)$. But $B[a]B^\top$ is a sparse Laplacian matrix, so we can use a specialized Laplacian solver that reduces the complexity to $O(|\mathcal{E}| \log^k |\mathcal{E}|)$ for some constant $k$ [33].

The bound on the number of iterations $k^*$ can be computed before any forward pass, since evaluating $h_\theta^{-1}$ does not require solving the IFN equations. But we can further simplify the bound by approximating $h_\theta^{-1}(0) = 0$, which is often justified because physical flow functions generally have a root at the origin. Using the fact that $(h_\theta^{-1})'(y) \leq d_{\min}^{-1} y$, we can then eliminate the dependence on $h_\theta^{-1}$:

$$k^* \leq 1 + \log\left(1 - \frac{d_{\min}}{d_{\max}}\right)\left(\log \epsilon - \log\left(||[a]^{-\frac{1}{2}} B^\dagger u||_2\right)\right)$$

## 2.2 Computing the Gradients

In order to train the flow function and any upstream models, it is necessary to backpropagate gradients through the IFN layer. We can perform this backward pass using implicit differentiation, and it turns out that the gradients of $\mathrm{FN}_{h,\theta}(G, u)$ with respect to the parameters $\theta$, $a$, and $u$ can also be computed using Algorithm 1, i.e., by writing the gradient as the output of an auxiliary implicit flow network.

**Theorem 2.4** (Gradients). *Consider an implicit flow network with parameters $d_{\min}$, $d_{\max}$, and $\theta$, with flow functions $h_\theta$. Suppose that the inputs $G = (\mathcal{V}, \mathcal{E}, a)$ and $u \in \mathbb{1}_{|\mathcal{V}|}^\perp$ are given, and let $B \in \{-1, 0, 1\}^{|\mathcal{V}| \times |\mathcal{E}|}$ be the signed incidence matrix of $G$. Let $f = \mathrm{FN}_{h,\theta}(G, u)$, and let $w$ be a scalar entry of $\theta$, $a$, or $u$. We can compute the derivatives $\frac{df}{dw}$ as follows.*

*Define a vector of flow functions $g : \mathbb{R}^{|\mathcal{E}|} \to \mathbb{R}^{|\mathcal{E}|}$ by*

$$g(\eta) = \mathcal{D}^{-1}\left(\eta - [a]^{-1}\frac{\partial v}{\partial w}\right), \quad \forall \eta \in \mathbb{R}^{|\mathcal{E}|} \tag{10}$$

*where $\mathcal{D} \in \mathbb{R}^{|\mathcal{E}| \times |\mathcal{E}|}$ is the diagonal matrix with entries*

$$\mathcal{D}_{ee} = \left.\frac{dh_\theta^{-1}(y_e)}{dy_e}\right|_{y_e = a_e^{-1} f_e}, \quad \forall e \in \mathcal{E} \tag{11}$$

*and $v = [a]h_\theta^{-1}([a]^{-1}f_{\mathrm{cyc}} + [a]^{-1}B^\dagger u)$. Then*

$$\frac{df}{dw} = \mathrm{FN}_{g,\cdot}(G, \mathbb{0}_n) + B^\dagger \frac{du}{dw} \tag{12}$$

*(We use the notation $\cdot$ in place of $\theta$, since $g$ has no trainable parameters.) Furthermore, the derivative constraint parameters $d_{\min}, d_{\max}$ from the original implicit flow network are valid for the new implicit flow network.*

In other words, to compute the gradient with respect to a parameter, we perform a single evaluation of the implicit flow network. In order to compute the derivatives with respect to some parameter or input $w$, we first evaluate the partial derivatives $\frac{\partial v}{\partial w}$ and the total derivatives $\frac{du}{dw}$. Then we construct the flow functions $g$ according to (10), and solve an implicit flow network to find $\frac{df}{dw}$ according to (12). It is easy to evaluate $\frac{du}{dw}$, but for convenience, we provide the values of $\frac{\partial v}{\partial w}$ below:

$$\frac{\partial v}{\partial \theta_i} = [a]\frac{dh_\theta^{-1}([a]^{-1}f)}{d\theta_i}, \qquad \frac{\partial v}{\partial a_e} = \mathrm{diag}\left(h_\theta^{-1}([a]^{-1}f) - [a]^{-1}\mathcal{D}f\right)_e, \qquad \frac{\partial v}{\partial u_i} = \left(\mathcal{D}B^\dagger\right)_i$$

## 3 Comparison with Optimization Models

Both of the state-of-the-art methods for flow estimation, from [3] and [4], use an optimization problem to predict flows. After a suitable transformation to incorporate external flow injections $u$, we can state this optimization problem as

$$\hat{f} = \operatorname*{argmin}_{f \in \mathbb{R}^{|\mathcal{E}|}}\left\{||f||_{2,[q]}^2 + \lambda^2||Bf - u||_2^2 \text{ s.t. } f_e = \tilde{f}_e, \ \forall \text{ labeled edges } e \in \mathcal{E}\right\} \tag{13}$$

where $\lambda > 0$, and $q > \mathbb{0}_m$ is some vector of edge weights. In [3], $q = \mathbb{1}_m$, while [4] allows $q$ to be the output of a neural network. IFN is not explicitly an optimization problem, but it can be cast as one that is similar to (13):

**Theorem 3.1** (Optimization Form of IFN). *Consider an IFN with flow function $h_\theta$. Suppose that the inputs $G = (\mathcal{V}, \mathcal{E}, a)$ and $u \in \mathbb{1}_{|\mathcal{V}|}^\perp$ are given, and let $B \in \{-1, 0, 1\}^{|\mathcal{V}| \times |\mathcal{E}|}$ be the signed incidence matrix of $G$. Then the IFN output can be stated as the solution of a convex optimization problem:*

$$\text{FN}_{h,\theta}(G, u) = \underset{f \in \mathbb{R}^m}{\arg\min} \left\{ \sum_{e \in \mathcal{E}} \int_0^{f_e} h_\theta^{-1}(a_e^{-1} z) \, dz \ s.t. \ Bf = u \right\} \tag{14}$$

Theorem 3.1 can be interpreted as a nonlinear generalization of the Thomson principle from electrical circuits theory [34]. Interestingly, the theorem sets up a direct comparison between IFN and the models in [3] and [4]. If the flow function $h_\theta$ is the identity map, then (14) can be simplified as

$$\text{FN}_{h,\theta}(G, u) = \underset{f \in \mathbb{R}^m}{\arg\min} \left\{ \sum_{e \in \mathcal{E}} ||f||_{2,[a]^{-1}}^2 \ \text{s.t.} \ Bf = u \right\} \tag{15}$$

Ignoring the constraints from labeled flows, we can interpret (13) as using a penalty method to approximate the output of an IFN with a linear flow function. Thus, we have three distinct differences between IFN and the optimization-based approaches. First, IFN allows for a nonlinear flow function, while [3] and [4] implicitly assume a linear CR. Second, IFN imposes flow conservation as a hard constraint rather than an approximate constraint (which is a limitation if $u$ is uncertain). Finally, IFN does not incorporate flow measurements directly; rather, the model exploits these measurements during training to learn the proper flow function (and train any upstream models for the IFN inputs), making it less sensitive to noise in the labeled flows.

## 4 Models for Flow Functions

In order to implement an IFN, it is necessary to parameterize its inverse flow function $h_\theta^{-1}$. Since the flow function is essentially a trainable activation function, i.e., a scalar nonlinearity, simple models are likely to be sufficient. The main difficulty with selecting a flow function is that its slope must be bounded by $d_{\min} \leq h_\theta'(y) \leq d_{\max}$ for all $y \in \mathbb{R}$. This section proposes a simple scalar nonlinearity that is guaranteed to respect arbitrary upper and lower bounds on its slope.

**Definition 4.1** (Derivative-Constrained Perceptron). Let $k \in \mathbb{Z}_{>0}$ be a hidden layer size, let $a, b, c \in \mathbb{R}^k$ be freely trainable parameters (encoded within the parameter vector $\theta$), and let $\sigma$ be a non-expansive activation. Let $p, q \geq 1$ such that $p^{-1} + q^{-1} = 1$, and let $\bar{d}_{\min} \leq \bar{d}_{\max} \in \mathbb{R}$. Then the *derivative-constrained perceptron* $\text{N}(x, \theta)$ is the 3-layer neural network defined by

$$\bar{c}(\theta) = \left( 1 - \frac{(||c||_p||a||_q - 1)_+}{||c||_p||a||_q} \right) c \tag{L1}$$

$$\text{N}_0(x, \theta) = \bar{c}^\top(\theta)\sigma(ax + b) \tag{L2}$$

$$\text{N}(x, \theta) = \left( \frac{\bar{d}_{\max} - \bar{d}_{\min}}{2} \right) \text{N}_0(x, \theta) + \left( \frac{\bar{d}_{\max} + \bar{d}_{\min}}{2} \right) x \tag{L3}$$

Intuitively, (L1) re-scales $c$ so that the perceptron in (L2) is guaranteed to be non-expansive in $x$. Then (L3) re-centers and re-scales the derivatives of the perceptron from the range $[-1, 1]$ to $[\bar{d}_{\min}, \bar{d}_{\max}]$.

**Theorem 4.2** (Derivative-Constrained Perceptron). *Let $\text{N}(x, \theta)$ be a derivative-constrained perceptron with $\bar{d}_{\min} \leq \bar{d}_{\max} \in \mathbb{R}$. Then for all parameter values $\theta$,*

$$\bar{d}_{\min} \leq \frac{d}{dx} \text{N}(x, \theta) \leq \bar{d}_{\max}, \quad \forall x \in \mathbb{R} \tag{16}$$

Note that the values $\bar{d}_{\min}, \bar{d}_{\max}$ in Definition 4.1 and Theorem 4.2 are distinct from the IFN parameters $d_{\min}, d_{\max}$. Since we parameterize the *inverse* flow function $h_\theta^{-1}$ in IFN, one shoudl set $\bar{d}_{\min} = d_{\max}^{-1}$ and $\bar{d}_{\max} = d_{\min}^{-1}$ to implement $h_\theta^{-1}$ with a derivative-constrained perceptron.

## 5 Numerical Experiments

We studied the transductive task of predicting unlabeled flows, given that some labeled flows in the same network are known. If the edges $\mathcal{E}$ are partitioned into a labeled set $\mathcal{E}_l$ and an unlabeled set $\mathcal{E}_u$,

the task is to predict the missing flows $\{f_e : e \in \mathcal{E}_u\}$ given the labeled flows $\{f_e : e \in \mathcal{E}_l\}$. For each network, we randomly selected a fraction of the edges to be labeled edges, and we trained IFN and baselines on the labeled edges. Then we evaluated the RMSE of the flows predicted for the unlabeled edges $\mathcal{E}_u$ to compute the testing error. See Appendix B in the supplementary material for full details. Code is available at https://github.com/KevinDalySmith/implicit-flow-networks.

## 5.1 Datasets

**AC Power** We selected 6 standard power network test cases. The first 4 test cases (IEEE-57, IEEE-118, IEEE-145, and IEEE-300) are synthetic transmission system test cases, while the remaining cases ACTIVSg200 and ACTIVSg500 are similar to the Illinois and South Carolina power grids, respectively [35]. Each test case contains the topology and electrical parameters of the power network, as well as baseline demands and power injections at each node. While branch resistances are typically small, we set them to zero to ensure lossless transmission lines. We used the MATPOWER toolbox [36] to solve the power flow equations, then recorded the active power flows on each branch ($f$), computed the net active power injections at each node ($u$), and selected relevant electrical parameters as edge attributes (series reactance, tap ratio, and voltage magnitude at the two incident nodes).

**Water Distribution** We selected 3 sample water distribution networks from the ASCE Task Committee on Research Databases for Water Distribution Systems database [37], representing municipal water distribution systems in Fairfield, CA, Bellingham, WA, and Harrisburg, PA. Each network contains the topology of the distribution system, as well as the characteristics of pipes and other network elements and nodal demands. We used the WNTR package [38] to compute the flow rates through each pipe ($f$), net inflow rate at each node ($u$), and edge weights associated with each pipe.

## 5.2 Models and Experiment Details

**IFN Architecture** In order to use the IFN layer to predict power flows, we created a two-layer model. The first layer estimates positive edge weights $a \in \mathbb{R}^{|\mathcal{E}|}$ according to $a_e = \exp(\mathrm{L}(z_e))$ for all $e \in \mathcal{E}$, where L is a linear module, and $z_e$ is the log-transformed vector of edge attributes. The second layer is an IFN. To predict water flows, we used an IFN layer alone, supplying the edge weights from the dataset as input (rather than learning them from other edge attributes). For both water and power, the IFN layer uses a derivative-constrained perceptron as the inverse flow function ($k = 128$, $p = q = \frac{1}{2}$) with a ReLU activation function. For power, we set $d_{\min} = 0.4$ and $d_{\max} = 2$; and for water, $d_{\min} = 0.2$ and $d_{\max} = 20$.

**Baselines** We compared the IFN model against four baselines. The minimum divergence method (*Div*) from [3] minimizes the nodal divergence $||Bf||_2^2$ and a regularization term $\lambda||f||_2^2$. The bilevel optimization methods from [4] replace the uniform regularizer with a weighted regularizer $||f||_{2,[q]}^2$, where $q$ is a vector of weights. In *Bil-MLP* and *Bil-GCN*, $q$ is the output of either a 2-layer MLP or GCN model with edge attributes as inputs (we use 64 nodes in each hidden layer with ReLU activations). In *Bil-True*, we specify $q$ as the reciprocal of the coefficient in the linearized CR for AC power networks, so that *Bil-True* approximates (15) with $a$ as the ground-truth edge weight. For water experiments, *Bil-True* uses the same edge weights as the IFN model.

All of the baselines assume that nodal divergence $Bf$ should be approximately zero, but nodes in power networks inject and withdraw power according to the supply / demand vector $u$, resulting in nonzero divergence. Thus, when we evaluate the baselines, we transform the power network into a divergence-free network by introducing a "source node", adding an edge from the source node to all nodes in $\mathcal{V}$, and treating the entries of $u$ as the flows along each corresponding virtual edge.

## 5.3 Results

Figure 2 reports the results for the AC power networks, and Figure 3 reports the results for water distribution systems. In both types of networks, the IFN model significantly outperforms the baselines on all of the networks when a small fraction of edges are labeled (less than 80% in power and less than 60% in water). While the other baselines tend to improve as more labeled edges are made available for training, IFN achieves near-optimal performance with as few as 10% of the edges labeled.

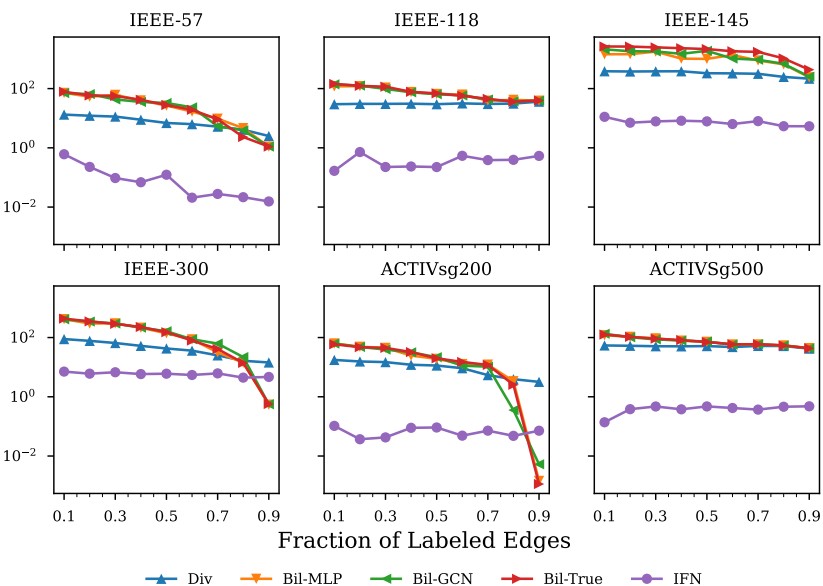

Figure 2: Results for missing flow prediction in AC power networks. Reported values are the RMSE (in units of MW) on the testing set, averaged across 10 trials. Note the vertical axis is in a log scale.

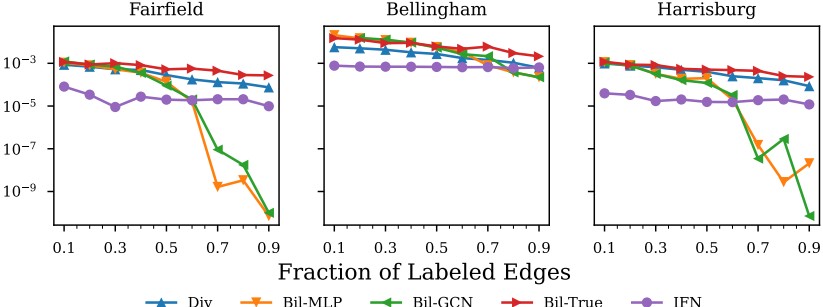

Figure 3: Results for missing flow prediction in water distribution systems. Reported values are the RMSE (in units of $m^3/s$) on the testing set, averaged across 10 trials.

## 6 Conclusion

In this paper, we have introduced an implicit model for network flows that incorporates physics through a conservation law and through the existence of a constitutive relationship between flows and nodal variables. We have demonstrated that a simple architecture using this model can learn to accurately predict active power flows in AC networks and water distribution systems. Future work may investigate more elaborate architectures using IFN as a layer, wherein the supply / demand vector, edge weights, or even the graph itself could be predicted from upstream models, and the flows themselves used for downstream tasks. Another interesting extension may be to extend our method to networks with higher-order interactions, i.e., hypergraphs [39] and simplicial complexes [40, 41].

IFN has some limitations that should also be addressed in future work. IFN assumes that the graph is undirected, which does not adequately model networks with unidirectional flows (e.g., traffic) or lossy flows (e.g., resistive power grids). IFN also assumes a CR that depends on the *difference* between nodal variables. This form appears frequently in physical systems, but in other network flow models (like Daganzo traffic models [42]), the CR has a more general dependence on the nodal variables. These limitations may be addressed with extensions of IFN's contraction algorithm.

## Acknowledgments and Disclosure of Funding

This work was supported by AFOSR under grant FA9550-22-1-0059, US Army ERDC under grant W912HZ-22-2-0010, and DTRA under grant HDTRA1-19-1-0017.

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
