# A Proofs

## A.1 Proof of Theorem 2.2

To prove statement 1, choose any $f_{\text{cyc}} \in \ker(B)$, let $y = [a]^{-1}f_{\text{cyc}} + [a]^{-1}B^\dagger u$ for brevity, and observe that

$$\left\| \frac{\partial T(f_{\text{cyc}})}{\partial f_{\text{cyc}}} \right\|_{2,[a]^{-1}} = \left\| [a]^{-\frac{1}{2}} P \left( I_m - d_{\min}[a]\frac{\partial h_\theta^{-1}(y)}{\partial y}[a]^{-1} \right) [a]^{\frac{1}{2}} \right\|_2$$

$$\leq \left\| [a]^{-\frac{1}{2}} P [a] \right\|_2 \left\| [a] \left( I_m - d_{\min}[a]\frac{\partial h_\theta^{-1}(y)}{\partial y}[a]^{-1} \right) [a]^{\frac{1}{2}} \right\|_2$$

$$= \left\| [a]^{-\frac{1}{2}} \left( I_m - d_{\min}[a]\frac{\partial h_\theta^{-1}(y)}{\partial y}[a]^{-1}[a]^{-1} \right) [a]^{\frac{1}{2}} \right\|_2$$

where $\||[a]^{-\frac{1}{2}} P[a]^{\frac{1}{2}}\||_2 = 1$ because $[a]^{-\frac{1}{2}} P[a]^{\frac{1}{2}}$ is a symmetric and idempotent matrix, i.e., an orthogonal projection. Then

$$\left\| \frac{\partial T(f_{\text{cyc}})}{\partial f_{\text{cyc}}} \right\|_{2,[a]^{-1}} = \max_{e \in \mathcal{E}} \left| 1 - d_{\min}(h_\theta^{-1})'(y_e) \right| \leq 1 - \frac{d_{\min}}{d_{\max}}$$

Hence

$$\text{Lip}(T) = \sup_{f_{\text{cyc}} \in \mathbb{R}^m} \left\| \frac{\partial T(f_{\text{cyc}})}{\partial f_{\text{cyc}}} \right\|_{2,[a]^{-1}} \leq 1 - \frac{d_{\min}}{d_{\max}} < 1$$

Then statement 2 follows from statement 1 and the Banach fixed point theorem. To prove statement 3, observe that $f_{\text{cyc}} = Pf_{\text{cyc}}$, so $f_{\text{cyc}} = T(f_{\text{cyc}})$ if and only if

$$P[a]h_\theta^{-1}\left([a]^{-1}f\right) = \mathbb{0}_m \tag{17}$$

where $f = f_{\text{cyc}} + B^\dagger u$. But $\ker(P[a]) = \text{Img}(B^\top)$, so (17) is equivalent to the existence of $x \in \mathbb{R}^n$ such that

$$h^{-1}([a]^{-1}f) = B^\top x \tag{18}$$

and (18) is equivalent to (4).

## A.2 Proof of Theorem 2.3

To prove statement 1, let $k \geq 0$ and consider iteration $k + 1$ of the loop. The iteration first defines $v = d_{\min}[a]h_\theta^{-1}\left([a]^{-1}f_{\text{cyc}}^{(k)} + [a]^{-1}f_{\text{cut}}\right)$ on line 6. Then on line 7,

$$w = \underset{w \in \mathbb{R}^n}{\text{argmin}} \left\{ ||B[a]B^\top w - Bv||_2 \right\} = \left(B[a]B^\top\right)^\dagger Bv$$

and line 8 defines

$$\Delta f_{\text{cyc}} = v - [a]B^\top w = \left( I_m - [a]B^\top \left(B[a]B^\top\right)^\dagger \right) v = Pv$$

Finally, on line 9,

$$f_{\text{cyc}}^{(k+1)} = f_{\text{cyc}}^{(k)} - Pv = f_{\text{cyc}}^{(k)} - d_{\min}P[a]h_\theta^{-1}\left([a]^{-1}f_{\text{cyc}}^{(k)} + [a]^{-1}f_{\text{cut}}\right)$$

A simple inductive argument shows that $f_{\text{cyc}}^{(k)} \in \ker(B)$. The base case $f_{\text{cyc}}^{(0)} = \mathbb{0}_m$ is trivial, for all $k' \geq 0$, line 9 ensures that $f_{\text{cyc}}^{(k'+1)} \in \ker(B)$ so long as $f_{\text{cyc}}^{(k')} \in \ker(B)$. Hence $f_{\text{cyc}}^{(k)} = Pf_{\text{cyc}}^{(k)}$, and we conclude that

$$f_{\text{cyc}}^{(k+1)} = P\left(f_{\text{cyc}}^{(k)} - d_{\min}[a]h_\theta^{-1}\left([a]^{-1}f_{\text{cyc}}^{(k)} + [a]^{-1}f_{\text{cut}}\right)\right) = T(f_{\text{cyc}}^{(k)})$$

To prove statement 2, recall from Theorem 2.2 that $\text{Lip}(T) \leq 1 - d_{\max}^{-1}d_{\min}$, which (together with statement 1) implies that, for all $k \geq 0$,

$$||f_{\text{cyc}}^{(k+1)} - f_{\text{cyc}}^{(k)}||_{2,[a]^{-1}} \leq \left(1 - \frac{d_{\min}}{d_{\max}}\right)^k ||f_{\text{cyc}}^{(1)} - f_{\text{cyc}}^{(0)}||_{2,[a]^{-1}}$$

$$= d_{\min}\left(1 - \frac{d_{\min}}{d_{\max}}\right)^k ||P[a]h_\theta^{-1}\left([a]^{-1}B^\dagger u\right)||_{2,[a]^{-1}}$$

$$= d_{\min}\left(1 - \frac{d_{\min}}{d_{\max}}\right)^k \rho$$

The algorithm terminates after iteration $k$ if and only if $||f_{\text{cyc}}^{(k)} - f_{\text{cyc}}^{(k-1)}||_{2,[a]^{-\frac{1}{2}}} \leq \epsilon$, so the algorithm will have terminated after $k^*$ iterations if

$$d_{\min}\left(1 - \frac{d_{\min}}{d_{\max}}\right)^{k^*-1} \rho \leq \epsilon$$

which is equivalent to

$$k^* \geq 1 + \frac{\log\left(d_{\min}^{-1}\rho^{-1}\epsilon\right)}{\log\left(1 - \frac{d_{\min}}{d_{\max}}\right)}$$

Finally, to prove statement 3, note that the algorithm terminates after iteration $k$ as soon as

$$||f_{\text{cyc}}^{(k)} - f_{\text{cyc}}^{(k-1)}||_{2,[a]^{-1}} \leq \epsilon$$

If $f_{\text{cyc}}$ is the true fixed point of $T$, then using a general property of contraction mappings,

$$||f_{\text{cyc}}^{(k)} - f_{\text{cyc}}||_{2,[a]^{-1}} \leq \frac{\text{Lip}(T)}{1 - \text{Lip}(T)}||f_{\text{cyc}}^{(k)} - f_{\text{cyc}}^{(k-1)}||_{2,[a]^{-1}}$$

$$\leq \left(\frac{d_{\max} - d_{\min}}{d_{\min}}\right)\epsilon$$

Therefore, the vector $f$ returned by the algorithm satisfies

$$||f - \text{FN}_{h,\theta}(G, u)||_{2,[a]^{-1}} = ||f_{\text{cyc}}^{(k)} - f_{\text{cyc}}||_{2,[a]^{-1}} \leq \left(\frac{d_{\max} - d_{\min}}{d_{\min}}\right)\epsilon$$

### A.3  Proof of Theorem 2.4

Let $v = [a]h_\theta^{-1}([a]^{-1}f_{\text{cyc}} + [a]^{-1}B^\dagger u)$. From Theorem 2.2, we can write $f = f_{\text{cyc}} + B^\dagger u$, where $f_{\text{cyc}}$ is the unique fixed point of $T$. Therefore $\frac{df}{dw} = \frac{df_{\text{cyc}}}{dw} + B^\dagger\frac{du}{dw}$, so the remainder of the proof is to show that $\frac{df_{\text{cyc}}}{dw} = \text{FN}_{g,\cdot}(G, \mathbb{0}_n)$.

Since $f_{\text{cyc}} = T(f_{\text{cyc}})$, and $Pf_{\text{cyc}} = f_{\text{cyc}}$, we have

$$f_{\text{cyc}} = P\left(f_{\text{cyc}} - d_{\min}v\right) = f_{\text{cyc}} - d_{\min}Pv$$

so an equivalent characterization of $f_{\text{cyc}}$ is the unique solution to the equations

$$Bf_{\text{cyc}} = \mathbb{0}_n$$
$$Pv = \mathbb{0}_m$$

Since

$$\frac{dv}{dw} = \frac{\partial v}{\partial w} + \frac{\partial v}{\partial f_{\text{cyc}}}\frac{df_{\text{cyc}}}{dw} = \frac{\partial v}{\partial w} + \mathcal{D}\frac{df_{\text{cyc}}}{dw}$$

then differentiating and factoring out $[a]$, we obtain

$$B\frac{df_{\text{cyc}}}{dw} = \mathbb{0}_n$$

$$P[a]\left([a]^{-1}\frac{\partial v}{\partial w} + [a]^{-1}\mathcal{D}\frac{df_{\text{cyc}}}{dw}\right) = \mathbb{0}_m$$

Since $\ker(P[a]) = \text{Img}(B^\top)$, there exists $x \in \mathbb{R}^n$ such that

$$[a]^{-1}\frac{\partial v}{\partial w} + [a]^{-1}\mathcal{D}\frac{df_{\text{cyc}}}{dw} = B^\top x$$

which we can re-write as

$$\frac{df_{\text{cyc}}}{dw} = [a]\mathcal{D}^{-1}\left(B^\top x - [a]^{-1}\frac{\partial v}{\partial w}\right) = [a]g(B^\top x)$$

Hence $\frac{df_{\text{cyc}}}{dw}$ is the solution to

$$B\frac{df_{\text{cyc}}}{dw} = \mathbb{0}_n \tag{19}$$

$$\frac{df_{\text{cyc}}}{dw} = [a]g(B^\top x) \tag{20}$$

which is identical to (3)–(4) with $\frac{df_{\text{cyc}}}{dw}$ in place of $f$, $\mathbb{0}_n$ in place of $u$, and $g$ in place of $h_\theta$. Furthermore, $g$ respects the same $d_{\min}, d_{\max}$ derivative constraints as $h_\theta$, since for each $e \in \mathcal{E}$,

$$g'_e(\eta_e) = \frac{1}{\mathcal{D}_{ee}} = \left.\frac{dh_\theta(y_e)}{dy_e}\right|_{y_e = a_e^{-1}f_e} \in [d_{\min}, d_{\max}]$$

It follows that $\frac{df_{\text{cyc}}}{dw}$ is the output of the implicit flow network with flow functions $g$ and parameters $d_{\min}, d_{\max}$, evaluated on the original graph $G$ and nodal demands $\mathbb{0}_n$.

## A.4  Proof of Theorem 3.1

Since $h_\theta^{-1}$ is increasing, the optimization problem in (14) has a convex cost function with linear constraints, so the KKT conditions are necessary and sufficient. Letting $x \in \mathbb{R}^m$ be a vector of Lagrange multipliers, the Lagrangian is

$$\mathcal{L} = \sum_{e \in \mathcal{E}} \int_0^{f_e} h_\theta^{-1}(a_e^{-1}z)\, dz - x^\top (Bf - u)$$

leading to the stationarity condition

$$\mathbb{0}_m^\top = \frac{\partial \mathcal{L}}{\partial f} = h_\theta^{-1}(f^\top[a]^{-1}) - x^\top B$$

which is equivalent to (4). Additionally, the primal constraint $Bf = u$ is equivalent to (3), so the minimizer of the optimization problem is identical to the output of the IFN.

## A.5  Proof of Theorem 4.2

Due to (L3), it is clear that the derivative bounds (16) hold if and only if

$$\left|\frac{d\text{N}_0(x, \theta)}{dx}\right| \leq 1, \qquad \forall x \in \mathbb{R} \tag{21}$$

For all $x, x' \in \mathbb{R}$, by Hölder's inequality,

$$|\text{N}_0(x, \theta) - \text{N}_0(x', \theta)| = \left|\bar{c}^\top(\theta)\left(\sigma(ax + b) - \sigma(ax' + b)\right)\right|$$
$$\leq ||\bar{c}(\theta)||_p\, ||\sigma(ax + b) - \sigma(ax' + b)||_q$$

Since $\sigma$ is non-expansive, its Lipschitz constant with respect to the $q$-norm is

$$\text{Lip}(\sigma) = \sup_{\eta \in \mathbb{R}^k}\left|\left|\frac{\partial\sigma(\eta)}{\partial\eta}\right|\right|_q = \sup_{\eta_0 \in \mathbb{R}}|\sigma'(\eta_0)| \leq 1$$

and thus

$$||\sigma(ax + b) - \sigma(ax' + b)||_q \leq ||a(x - x')||_q \leq ||a||_q|x - x'|$$

| Test Case | MATPOWER Case Name | $|\mathcal{V}|$ | $|\mathcal{E}|$ |
|---|---|---|---|
| IEEE-57 | case57 | 57 | 135 |
| IEEE-118 | case118 | 118 | 297 |
| IEEE-145 | case145 | 145 | 567 |
| IEEE-300 | case300 | 300 | 709 |
| ACTIVSg200 | case_ACTIVSg200 | 200 | 445 |
| ACTIVSg500 | case_ACTIVSg500 | 500 | 1084 |

Table 2: MATPOWER test case details.

Furthermore, by (L1),

$$||\bar{c}(\theta)||_p||a||_q = \left(1 - \frac{(||c||_p||a||_q - 1)_+}{||c||_p||a||_q}\right)||c||_p||a||_q$$
$$= ||c||_p||a||_q - (||c||_p||a||_q - 1)_+$$
$$= \min\{1, ||c||_p||a||_q\}$$

so that

$$|\mathrm{N}_0(x, \theta) - \mathrm{N}_0(x', \theta)| \le \min\{1, ||c||_p||a||_q\}|x - x'| \le |x - x'|$$

for all $x, x'$. Hence (21) is satisfied.

## B   Experiment Details

### B.1   AC Power Datasets

We created datasets from 6 AC power network test cases. Each dataset that we created represents a snapshot of an AC power network in its steady state, consisting of four components: the network topology (as an oriented, undirected graph), four attributes on each edge (voltage magnitude at the two incident nodes, series reactance, and tap ratio), the net power injection at each node, and the active power flow through each branch.

**Original Data**   We generated our datasets using MATPOWER, an open-source toolbox for power system simulation in MATLAB [36]. The toolbox includes many standard test cases, which contain a network topology and tables of electrical and economic parameters for each bus (node), branch (edge), and generator. We selected 6 test cases, listed in Table 2. The raw data files for these test cases are available from the MATPOWER source[3], and details on the test case file format are contained in Appendix B of the user manual[4].

**Data Generation**   After loading each test case into MATPOWER, we performed the following two modifications of the network parameters:

1. We set branch resistances (column 3 in the branch data table) to zero, so that transmission lines in the system are lossless. This step was necessary because IFN is limited to undirected graphs, while lossy lines are more appropriately modeled with a pair of directed edges, since the power injected at one endpoint does not equal the power withdrawn from the other endpoint. Fortunately, branch resistances are typically small before this modification.

2. We replaced any negative series reactances (column 4 in the branch data table) with a positive value, chosen as the median of the positive series reactances in the same network. We performed this modification because negative series reactances results in *decreasing* constitutive relationships on the corresponding edges, whereas IFN assumes that the constitutive relationship is increasing. This modification only affected two networks: IEEE-145, in which 24 (4.2%) of the branches were assigned a series reactance of 0.2306; and IEEE-300, in which 1 (0.1%) of the branches was assigned a series reactance of 0.059.

We then computed the resulting power flows using the `runpf` function and recorded the results.

---

[3]https://github.com/MATPOWER/matpower/tree/master/data
[4]https://matpower.org/docs/MATPOWER-manual.pdf

| Test Case | $|\mathcal{V}|$ | $|\mathcal{E}|$ |
|---|---|---|
| Fairfield | 111 | 125 |
| Bellingham | 121 | 162 |
| Harrisburg | 261 | 286 |

Table 3: Water distribution network details.

**Pre-Processing**   Finally, we converted the results from the MATPOWER simulation into a PyTorch Geometric data object, with the following attributes:

- `edge_index`, the edge index tensor, containing the topology from the test case.

- `x`, a tensor of net active power injections at each node, which has the property that $\mathbb{1}_n^\top x = 0$. (This tensor is identical to the supply / demand vector $u$ in the paper.)

- `edge_attr`, a tensor of four relevant attributes for each edge: the voltage magnitudes at the two incident nodes, the series reactance, and the tap ratio.

- `f_true`, the tensor of active power flows on each edge simulated by MATPOWER.

The net active power injections at each node are computed according to

$$u_i = \mathrm{PG}_i - \mathrm{PD}_i - \mathrm{GS}_i \mathrm{VM}_i^2$$

where $\mathrm{PG}_i$ is active power generated at $i$, $\mathrm{PD}_i$ is active power demanded, $\mathrm{GS}_i$ is shunt conductance, and $\mathrm{VM}_i$ is the voltage magnitude.

## B.2   Water Distribution Dataset

We created 3 datasets representing snapshots of municipal water distribution networks in their steady state, consisting of four components: the network topology (as an oriented, undirected graph), weights for each edge, the net inflow rates at each node, and the flow rate through each pipe.

**Original Data**   Each of the datasets is based on a network from the ASCE Task Committee on Research Databases for Water Distribution Systems database [37]. Networks in this database contain a distribution network topology and tables of hydraulic parameters and operating characteristics for each node, pipe (edge), pump, reservoir, and storage tank in the network. We selected 3 networks, listed in Table 3 and plotted in Figure 4. The raw data files are available online[5].

**Data Generation and Preprocessing**   We loaded each network INP file into WNTR and ran the WNTR simulator with a hydraulic accuracy of $10^{-8}$. We then converted the results into a PyTorch Geometric data object, with the following attributes:

- `edge_index`, the edge index tensor, containing the topology from the test case.

- `x`, a tensor of net inflows at each node, which has the property that $\mathbb{1}_n^\top x = 0$.

- `edge_attr`, a tensor of three relevant attributes for each edge: the pipe length, pipe diameter, and pipe roughness coefficient.

- `f_true`, the tensor of flow rates through each pipe simulated by WNTR.

Edge weights are computed according to the formula

$$a_e = (0.27855)C_e D_e^{2.63} L_e^{-0.54} \tag{22}$$

where $C_e$ is the roughness coefficient (unitless), $D_e$ is the diameter (meters), and $L_e$ is the pipe length (meters) [6].

---

[5] http://www.uky.edu/WDST/index.html
[6] https://wntr.readthedocs.io/en/latest/hydraulics.html

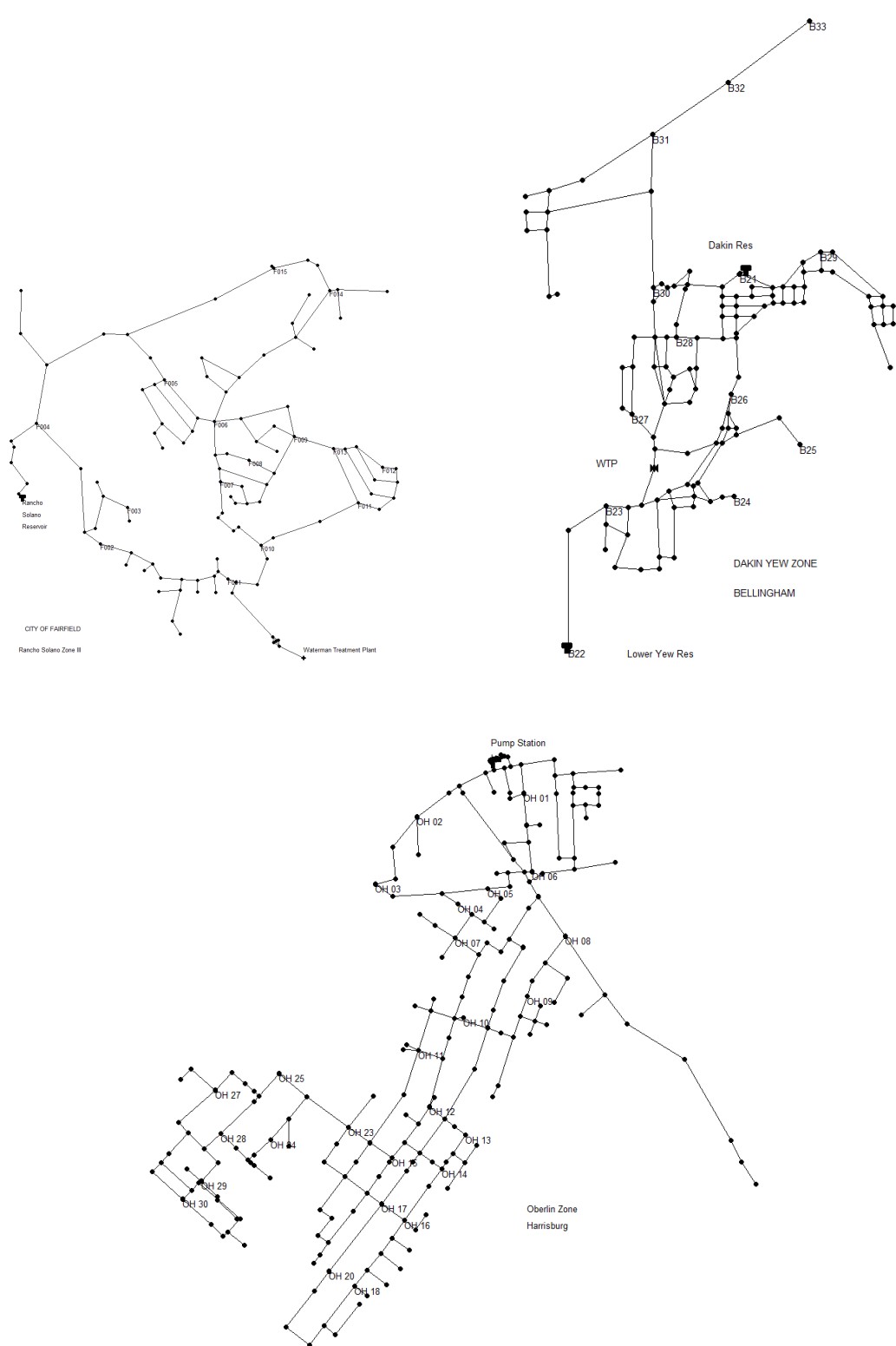

Figure 4: Network maps of the three water distribution systems: Fairfield (upper left), Bellingham (upper right), and Harrisburg (bottom).

## B.3 Details on IFN

Our IFN implementation uses Algorithm 1 to compute the layer's forward pass. We set the maximum number of iterations in this algorithm to 100, with a tolerance of $\epsilon = 10^{-2}$ for power and $\epsilon = 10^{-4}$ for water. With the release of PyTorch 1.11.0, the `torch.linalg.lstsq` method[7] now supports automatic differentiation, allowing PyTorch to automatically backpropagate through the Algorithm 1 iterations, instead of using Theorem 2.4. We found that Algorithm 1 terminated with a small enough number of iterations that automatic differentiation was faster, so we opted to use this rather than the method from Theorem 2.4. We trained the IFN models to minimize the RMSE loss function by minimizing the RMSE loss function

$$\ell_{\text{rmse}} = \sqrt{\frac{1}{|\mathcal{E}_l|} \sum_{e \in \mathcal{E}} \left(f_e - \text{FN}_{h,\theta}(G, u)_e\right)^2}$$

## B.4 Details on Baselines

We implemented all of the baselines by adapting Silva's code[8] from [4], refactoring some utility functions to decrease runtime. Following [4], we perform the following two data normalization steps:

1. negative flows are converted into positive flows by flipping the orientation of the corresponding edges and replacing the entries of `f_true` with their absolute value, and
2. flows are proportionally normalized to the range $[0, 1]$ within each network.

After training with the normalized flows and computing the missing flow predictions, the predictions are denormalized before computing the testing RMSE.

**Div** The minimizing divergence baseline from [3] has a single hyperparameter, $\lambda$, from the regularization term $\lambda^2 ||f||_2^2$ in the loss function. We set $\lambda = 0.1$ for all networks and fractions of labeled edges by hand-tuning the parameter to the proper order of magnitude.

**Bilevel Baselines** All three of the bilevel baselines (Bil-MLP, Bil-GCN, and Bil-True) have several hyperparameters related to the bilevel optimization algorithm. For most of these parameters, we use the same settings as [4]: the number of iterations for the inner optimization problem is 300 during training and 3000 during evaluation, and the number of k-fold cross validation folds is 10; however, we increased the number of iterations of the outer optimization problem from 10 to 100, with an early stopping interval of 10, to ensure that the outer optimization problem was given sufficient time to converge. As with [4], we used a 2-layer MLP and GCN in Bil-MLP and Bil-GCN, respectively, but we increased the size of the hidden layer to 64.

**Bil-True** Like IFN, the baselines Bil-MLP and Bil-GCN train a model to predict edge weights from side information (if we interpret $\mathcal{Q}$ as a diagonal matrix of edge weights). We devised Bil-True as a third baseline to use the "ground-truth edge weights" instead of training a model. For water experiments, these ground-truth edge weights are given by (22). For the power experiments, we compute these edges weights from the AC active power flow equation: in a lossless AC power grid, active power flows $f_{ij}$ on each edge $\{i, j\} \in \mathcal{E}$ are given by

$$f_{ij} = \frac{v_i v_j}{x_{ij} \tau_{ij}} \sin(\theta_i - \theta_j) \approx \frac{v_i v_j}{x_{ij} \tau_{ij}} (\theta_i - \theta_j) \tag{23}$$

where $v_i, v_j$ are the voltage magnitudes on incident nodes, $x_{ij}$ is the series reactance, $\tau_{ij}$ is the tap ratio, and $\theta_i, \theta_j$ are the incident voltage angles. Since (23) is the constitutive relationship for AC power networks, examining its linear approximation in light of (15) suggests using $x_{ij}\tau_{ij}/v_i v_j$ as the regularizer weight on $f_{ij}$.

## B.5 Details on Training

We trained all models in a Google Colab notebook, using the Adam optimizer. We used an initial learning rate of 0.01, which we found to have good performance for all of the models. Training was terminated when the training loss had not decreased for 10 epochs.

---

[7] https://pytorch.org/docs/stable/generated/torch.linalg.lstsq.html
[8] https://openreview.net/forum?id=l0V53bErniB