# OpenReview forum: "Physics-Informed Implicit Representations of Equilibrium Network Flows"
_NeurIPS.cc/2022/Conference — NeurIPS 2022 Accept_

### Official Review · Reviewer_dwJZ · 2022-07-02

**Rating:** 5
**Confidence:** 3
**Soundness:** 2 fair
**Presentation:** 3 good
**Contribution:** 3 good

**Summary:**

This paper incorporates two physical laws, namely conservation of mass and the existence of a constitutive relationship between edge flows and nodal states, into an implicit neural network layer to solve the estimation problem of unlabeled edge flows.

Rating raised from 4 to 5 after revision.

**Questions:**

N/A

**Limitations:**

Yes. The authors address the limitations of the implicit neural network layer that needs to be improved in the future.

**Strengths And Weaknesses:**

Theory:
1. Applying these two physical laws to flow prediction seems natural and reasonable. I’m not sure if others have made similar attempts.
2. The mathematics in the paper is cool, although it might be a little hard to follow.

Presentation:
1. How the physical laws are incorporated is not highlighted in the paper and is hard to find. Section 3.1 is too far away from Equations 1 and 2.  After the revision, this point has been demonstrated more clearly.
2. The physical laws incorporated are not elaborated.
3. Figures sometimes are too far away from the corresponding text. Solved after revision.

Experiment: Although the paper lists a few potential applications of flow prediction, the experiment focuses only on AC power networks. Thus, only one example of the second physical law, which is Ohm’s law, is examined. More experiments are included after the revision.

---

> ### Author Response · Authors · 2022-08-02
> **Response to Reviewer dwJZ**
>
> Thank you for reviewing our paper and for your insightful comments!
>
> ## Strengths and Weaknesses
>
> > The mathematics in the paper is cool, although it might be a little hard to follow.
>
> We're glad you enjoyed the math! We have made some slight modifications to improve the readability of key equations. We added new equations (1) and (2), which state the main IFN equations (3) and (4) (previously (1) and (2) in the original document) in a simpler component-wise form. We have also added a simpler approximation for the bound $k^*$ on the number of Algorithm 1 iterations. Please let us know if there are any other equations, proof steps, symbols, etc. in particular that deserve more elaboration or explanation in the text.
>
> > (1) How the physical laws are incorporated is not highlighted in the paper and is hard to find. Section 3.1 is too far away from Equations 1 and 2. (2) The physical laws incorporated are not elaborated.
>
> Thank you for these comments; we agree that the original document did not make explicit where the physical laws are incorporated in IFN. We have re-written the beginning of Section 2 to clarify the introduction of IFN, highlight how the two physical laws are incorporated in the model, and provide earlier intuition for the relevant variables. We have also moved the content of Section 3.1 to the start of Section 2, and we added a sentence after Definition 2.1 explicitly linking equations (1) and (2) (now (3) and (4)) to the conservation law and the constitutive relationship. Hopefully the incorporation of the physical laws is more clear with these changes.
>
> > Figures sometimes are too far away from the corresponding text.
>
> Thank you for pointing this out; we have ensured that Figure 1 (and its reference in the text) is now on the same page as the text describing the flow space decomposition. With the new water network plots, Figures 2 and 3 are now on the page following the corresponding text, but unfortunately we don't see a good way to resolve this.
>
> > Although the paper lists a few potential applications of flow prediction, the experiment focuses only on AC power networks. Thus, only one example of the second physical law, which is Ohm’s law, is examined.
>
> We have added results from water distribution networks in the latest revision, providing a second example of the second physical law.

---

### Official Review · Reviewer_k7us · 2022-07-11

**Rating:** 7
**Confidence:** 4
**Soundness:** 3 good
**Presentation:** 3 good
**Contribution:** 3 good

**Summary:**

The paper introduces an implicit neural network to compute (nonlinear) network flows, extending previous works that predict flows based on a divergence minimization principle.

**Questions:**

Why do the authors only consider primarily power flow problems in their evaluation?

It appears that one of the strengths of the proposed framework would be that it can be adjusted to different constitutive laws, covering other scenarios as well. It would be nice to see an evaluation of this. Several further datasets (traffic, etc.) are available from Ref [3] here:
https://github.com/000Justin000/ssl_edge/tree/master/data

I think a somewhat broader discussion of related literature would be of interest to the reader.

In the context of solving power flows with (graph) neural networks, see. e.g.:
Böttcher, Luis, et al. "Solving AC Power Flow with Graph Neural Networks under Realistic Constraints." arXiv preprint arXiv:2204.07000 (2022).
There are also several neural networks for edge flow predictions:

Roddenberry, T. Mitchell, Nicholas Glaze, and Santiago Segarra. "Principled simplicial neural networks for trajectory prediction." International Conference on Machine Learning. PMLR, 2021.
Ebli, Stefania, Michaël Defferrard, and Gard Spreemann. "Simplicial Neural Networks." NeurIPS 2020 Workshop on Topological Data Analysis and Beyond. 2020.

Finally, related works can also be found in the graph signal processing literature, e.g.,
Yang, Maosheng, et al. "Simplicial Convolutional Filters." arXiv preprint arXiv:2201.11720 (2022).



**Limitations:**

n/a -- limitations are not discussed

**Strengths And Weaknesses:**

Strengths:
- well written paper
- clear theoretical characterization of results

Weaknesses
- somewhat limited numerical evaluation (only power flow datasets)
- related literature could be improved

---

> ### Author Response · Authors · 2022-08-02
> **Response to Reviewer k7us**
>
> Thank you for reviewing our paper and for your insightful comments and suggestions!
>
> ## Questions
>
> > Why do the authors only consider primarily power flow problems in their evaluation? It appears that one of the strengths of the proposed framework would be that it can be adjusted to different constitutive laws, covering other scenarios as well. It would be nice to see an evaluation of this. Several further datasets (traffic, etc.) are available from Ref [3]...
>
> In the revised document, we have added experiments from water distribution systems to supplement the results from AC power networks (the new results are in Figure 3). Thanks for sharing these datasets! We are particularly interested in predicting traffic flows in future work---IFN in its current form is restricted to undirected graphs, but we believe an extension to directed networks is possible (albeit nontrivial), and traffic flow prediction would be a primary motivation for this extension.
>
> > I think a somewhat broader discussion of related literature would be of interest to the reader...
>
> Thank you for providing these references! Böttcher's paper in particular is very relevant to our work; we have added a reference in our paragraph on network flow estimation literature. We have also added the references on simplicial complexes in our conclusion, where we had mentioned hypergraphs as a possible future direction.

---

> > ### Comment · Reviewer_k7us · 2022-08-08
> > **thanks for the comments**
> >
> > I thank the authors for their comments. My rating remains as is on accept.

---

### Official Review · Reviewer_XX9M · 2022-07-11

**Rating:** 5
**Confidence:** 2
**Soundness:** 3 good
**Presentation:** 2 fair
**Contribution:** 3 good

**Summary:**

The submission proposes a solution to solve a conservation law and the existence of a constitutive relationship via implicit function. They shows that this model can estimate flows in AC networks.

**Questions:**

Is this work still working when the number of nodes of the graph increases or decreases?

**Limitations:**

This work is limited to the undirected graph.

**Strengths And Weaknesses:**

Strengths:

- It improved the previous general models predicting network flows not only by considering conservation law but also by considering a constitutive relationship.
- This work clearly and accurately incorporates the constitutive relationship in the implicit model.
- This work shows detailed descriptions for the forward and backward pass, where a contraction algorithm is used for evaluation and backward pass.

Weaknesses:

- This work is limited to an undirected graph.

---

> ### Author Response · Authors · 2022-08-02
> **Response to Reviewer XX9M**
>
> Thank you for reviewing our paper and for your insightful comments!
>
> ## Questions
>
> > Is this work still working when the number of nodes of the graph increases or decreases?
>
> There is no theoretical limitation on the number of nodes (as long as the graph has at least one edge). Of course, the computational complexities of the forward and backward passes increase with network size (more directly in the number of edges than the number of nodes); this dependency can be seen in $k^*$ in (6) (now (8)), since $\rho$ tends to increase with the number of edges. Our AC power experiments involve networks with ~1k edges.

---

### Official Review · Reviewer_JgzR · 2022-07-11

**Rating:** 5
**Confidence:** 4
**Soundness:** 4 excellent
**Presentation:** 3 good
**Contribution:** 3 good

**Summary:**

IFN provides an implicit neural network layer, which can be used to predict unlabeled edge flows (f) from nodal supply and demand (u), which can be formulated as a nonlinear inverse problem. It also takes advantage of the physics of the flow network, namely conservation of mass at nodal points and constitutive relationship (Eqns 1 and 2) to train h_\theta. Analysis is given for the convergence and existence of the IFN. To demonstrate, examples in the context of AC power networks are given.


**Questions:**

Minor suggestion: maybe theorem 2.2 can be expressed as a corollary right after theorem 2.3. Of course. authors can still mention the well-posedness of IFN before section 2.1, whose proof will be provided later.

Can we have an estimation of k* a-priori? Since it is a function of h_{\theta}, it seems like it can be only found after the training. But it might be nice to have an approximation of the number of iterations.

Any comments for the time-dependent flow networks? Please note that for various physics-informed networks, a major limitations is capturing the time-dependencies, so it's a non-trivial task (but feasible).

The authors clearly show how nonlinearities in equation 2 are captured, but seems like equation 1 needs to be linear. This may be a limiting factor. Any comments for a nonlinear version of equation 1 (for example conservation of momentum).

Although IFN does not include measurements; can you consider it an advantage? Let's say constitutive law is not completely understood. So at the time of training, h_\theta and f are evaluated based on the current model but such model may bifurcate or change in action. Since IFN does not take advantage of measurements directly, it may be prone to error in such circumstance. Any comments on robustness of IFN?



**Limitations:**

Yes.

**Strengths And Weaknesses:**

Strengths:
The method to enforce physical laws of eqns 1 and 2 into the training is elegant. It comes with guarantees on the existence of the solution (contraction of T, which also makes the inverse problem a well-posed one), algorithm with clear mathematical foundation (with a finite number of iteration for convergence k*), and convergence analysis on FN (forward pass).
The authors further compute gradients required for training in manner specific to IFN.
The distinction with optimization-based approaches is clear.

Weaknesses:
The implicit flow network represented by a neural network is not new; but addition of physical laws (conservation and constitutive) is novel. Authors also develop a nice method to achieve this. But although they try to generalize this for a verity of physical systems in section 3.1, I don't think it is clear how can one extend the framework for more complicated networks such as traffic flow or collective system of rigid bodies. Furthermore, seems like the method can only be applied to steady-state systems which is a very serious limitations. I think such limitations should reflect in the title of the paper as the current one may sound too generic.

---

> ### Author Response · Authors · 2022-08-02
> **Response to Reviewer JgzR (Strengths and Weaknesses)**
>
> Thank you for reviewing our paper and for your insightful comments and suggestions!
>
> ## Strengths and Weaknesses
>
> > But although they try to generalize this for a verity of physical systems in section 3.1, I don't think it is clear how can one extend the framework for more complicated networks such as traffic flow or collective system of rigid bodies.
>
> Indeed IFN does not apply to traffic flow, since IFN's contraction algorithm is restricted to undirected graphs, while traffic networks are directed. Extending the contraction algorithm to directed networks is indeed possible, although such an extension would be in the realm of future work.
>
> Predicting equilibria for systems of rigid bodies is another interesting direction. Solving statics problems with IFN would require an extension of the framework to higher-dimensional flows and latent nodal variables, which is also possible in principle. Given a network of rigid bodies, we would represent interaction force vectors as 3D "flows" on each edge, which are related to nodal coordinates by a 3D constitutive relationship (e.g., Hooke's law). Extending the contraction algorithm to higher dimensions is not trivial, so we leave this to future work, but thank you for pointing out this compelling direction.
>
> > Furthermore, seems like the method can only be applied to steady-state systems which is a very serious limitations. I think such limitations should reflect in the title of the paper as the current one may sound too generic.
>
> Thank you for this suggestion on how to clarify the scope of our method. We have changed the end of our title to "equilibrium network flows" and will ask the ACs about how to make this change official.

---

> ### Author Response · Authors · 2022-08-02
> **Response to Reviewer JgzR (Questions)**
>
> ## Questions
>
> > Minor suggestion: maybe theorem 2.2 can be expressed as a corollary right after theorem 2.3. Of course. authors can still mention the well-posedness of IFN before section 2.1, whose proof will be provided later.
>
> Thank you for this suggestion. We have removed Theorem 2.2 and instead state the well-posedness of IFN inside of Theorem 2.3 (now Theorem 2.2).
>
> > Can we have an estimation of k* a-priori? Since it is a function of h_{\theta}, it seems like it can be only found after the training. But it might be nice to have an approximation of the number of iterations.
>
> Yes! Typically $h_{\theta}^{-1}(0)$ is close to zero, since $h(0) = 0$ in symmetric flow networks (e.g., every example in Table 1). With this approximation, we can upper bound $k^*$ without evaluating $h_{\theta}^{-1}$. We have added a paragraph about this upper bound following (what is now) Theorem 2.3. However, note that one does not need to wait for training to finish to evaluate $k^*$ exactly, since $h_{\theta}^{-1}$ is the current estimate that IFN will use when evaluating the forward / backward passes.
>
> > Any comments for the time-dependent flow networks? Please note that for various physics-informed networks, a major limitations is capturing the time-dependencies, so it's a non-trivial task (but feasible).
>
> There are two time scales to consider: short time scales, when transient edge (or node) dynamics are relevant; and long time scales, when the system has settled to its steady state. IFN is probably not the right method for predicting fast dynamics, because constitutive relationships (one of the two physical laws IFN is derived from) are themselves equilibrium equations. For example, the sine relationship in AC power networks emerges after electromagnetic transients on the transmission lines have decayed. Learning the differential equations that govern these transient dynamics (system identification) is possible, but it requires data from additional state variables and / or strong assumptions on these differential equations.
>
> But IFN is well-suited for predictions at long time scales, i.e., modeling how steady states change over time, since conservation laws and constitutive relationships are generally time-invariant. Instead of predicting missing flows within a single snapshot, one could train IFN on historical snapshots to predict future flows (corresponding to future inflow vectors $u$ and future edge weights). This is a simpler task than predicting fast dynamics, but it is still well-motivated, since many operational decisions are made at this longer time scale (e.g., hours or days). Many popular simulation tools for power (MATPOWER, GridLAB-D, PyPSA) and water distribution (EPANET, WNTR) only compute flow steady-states instead of the fast dynamics.
>
> > The authors clearly show how nonlinearities in equation 2 are captured, but seems like equation 1 needs to be linear. This may be a limiting factor. Any comments for a nonlinear version of equation 1 (for example conservation of momentum).
>
> The linearity of (1) (now (3) in the revised document) comes from the underlying physics. We have added a new equation (now (1)) to make this point somewhat more explicit: the equation states that net inflows must balance against net outflows at each node, which is intrinsically a linear constraint. In theory, this linear conservation could apply to a property derived from the flows rather than the flows directly, leading to a nonlinear equation, but we don't have any examples of this in mind.
>
> It is not clear to us that momentum is an example of nonlinear conservation. If we have a network of rigid bodies, then conservation of momentum is a network-level constraint, whereas (1) is a nodal constraint. If we interpret edge flows as forces, then the time derivative of the network's total momentum is $\sum_i u_i$, so conservation of momentum is a constraint on $u$ rather than a nodal balance condition.
>
> > Although IFN does not include measurements; can you consider it an advantage? Let's say constitutive law is not completely understood. So at the time of training, h_\theta and f are evaluated based on the current model but such model may bifurcate or change in action. Since IFN does not take advantage of measurements directly, it may be prone to error in such circumstance. Any comments on robustness of IFN?
>
> Would you be able to clarify the proposed scenario? The output of IFN will only change if its flow function is changed (e.g., through training) or if the inputs are changed (e.g., through training or inductive evaluation). So long as IFN is optimized against a loss function that incorporates the measured flows, training should act against such an error. If the IFN output changes because the inputs change, i.e. it is evaluated on a different network, the previously measured flows from the old network would not be very useful. If measurements from the new network are also available, these can be treated as training data.

---

> > ### Comment · Reviewer_JgzR · 2022-08-08
> > **Reviewer JgzR response II**
> >
> > Thanks to the authors for their time and effort to improve the paper. I think authors are clear about the limitations of the method and in the revised version they improved upon this.
> > Regarding my last question (IFN and measurements), what I meant in short, is how do you see the usage of IFN in the face of uncertainties? Two types of uncertainties could be about the process (physical laws), and data (measurement noise). This probably falls out of scope of this paper, but eventually IFN-type works may face such challenges so it'd be helpful for readers to see your position on uncertainties and their quantifications.
> > My other concerns are responded.

---

> > > ### Author Response · Authors · 2022-08-09
> > > **Response II to Reviewer JgzR**
> > >
> > > Thank you for the clarification!
> > >
> > > Regarding process uncertainty, we would like to point out how we only assume the existence of a constitutive relation as in Eq. (2), without characterizing it explicitly. Since no additional constraint is enforced, IFN does not try to reconstruct flows according to a predetermined function, but simply estimates flows that obey an unknown constitutive equation. This feature makes IFN well suited to applications with process uncertainty, and is also particularly relevant to domains where constitutive relations are known only empirically, as it is the case for water distribution networks.
> > >
> > > Robustness to measurement uncertainty is an interesting question not only for IFN, but for implicit models in general.
> > > One could analyze the robustness of IFN against uncertainty in inputs (the $u$ vector and edge weights) through the Lipschitz constant of the map from the input to the predicted flow. (For example, Theorem 3 in ref. [14] provides a way to compute this.)
> > > Robustness to noise in edge flow measurements is trickier to analyze theoretically, since IFN incorporates this information only through training; we plan on assessing the impact of this particular measurement noise on IFN's performance relative to the optimization approaches numerically.

---

### Author Response · Authors · 2022-08-02
**Summary of Revisions (8/1)**

We would like to thank the reviewers for their efforts and helpful suggestions to improve our paper. In response to the reviewers' suggestions and questions, we have uploaded a revised manuscript, with the following updates:

## Major changes:

 - We have added experiments from water distribution systems to supplement the results from AC power networks. The new results are in Figure 3 and provide additional evidence of IFN's superior performance when most edges are unlabeled. Details on the water dataset were added to Appendix B.

 - We have re-written the beginning of Section 2 (before Section 2.1) to clarify the introduction of IFN, highlight how the two physical laws are incorporated in the model, and provide earlier intuition for the relevant variables. Per the request of Reviewer dwJZ, we have moved the content of Section 3.1 (examples of real-world constitutive relationships) to the start of Section 2. We hope that writing out the physical laws component-wise in equations (1) and (2), as well as having some immediate concrete examples, will make the physical laws in IFN more explicit.

## Minor changes:

 - As requested by Reviewer JgzR, we have changed the end of our title to "equilibrium network flows" to clarify the application to steady-state flows. We will ask the ACs about officially changing the title.

 - Per the suggestion of Reviewer JgzR, we have removed Theorem 2.2 (the well-posedness of IFN) and moved this statement to Theorem 2.3 (now Theorem 2.2).

 - We corrected a minor error in Theorems 2.3 and 2.4 (now 2.2 and 2.3). In the previous document, we had erroneously referred to the $\ell_{2, [a]^{-1/2}}$ norm, when we were actually using the $\ell_{2, [a]^{-1}}$ norm. This correction has slightly changed the statements of Theorems 2.3 (statement 1) and 2.4 (the definition of $\rho$) and Algorithm 1 (line 5).

 - We have added additional references suggested by Reviewer k7us.

 - Additionally, we have fixed several typos, made minor edits for concision, and moved some experiment details to the appendix to save space.

---

### Meta-Review · Area_Chair_CuT3 · 2022-08-29

**Recommendation:** Accept
**Confidence:** Less certain

**Metareview:**

This paper proposes Implicit Flow Network to estimate flow over network edges under certain physical constraints (conversation law and constitutive relationship such as Ohm’s law). Most reviewers agreed that the proposed idea of augmenting IFNs with physical constraints is novel and interesting, and experimental validation is sufficiently convincing.

**Award:**

No

---

### Decision · Program_Chairs · 2022-09-14

Accept